# Characteristics of Probiotic Preparations and Their Applications

**DOI:** 10.3390/foods11162472

**Published:** 2022-08-16

**Authors:** Guangqiang Wang, Yunhui Chen, Yongjun Xia, Xin Song, Lianzhong Ai

**Affiliations:** Shanghai Engineering Research Center of Food Microbiology, School of Health Science and Engineering, University of Shanghai for Science and Technology, Shanghai 200093, China

**Keywords:** probiotics, survival rate, targeted delivery system, dosage form, disease treatment

## Abstract

The probiotics market is one of the fastest growing segments of the food industry as there is growing scientific evidence of the positive health effects of probiotics on consumers. Currently, there are various forms of probiotic products and they can be categorized according to dosage form and the site of action. To increase the effectiveness of probiotic preparations, they need to be specifically designed so they can target different sites, such as the oral, upper respiratory or gastrointestinal tracts. Here we review the characteristics of different dosage forms of probiotics and discuss methods to improve their bioavailability in detail, in the hope that this article will provide a reference for the development of probiotic products.

## 1. Introduction

In recent years, there has been increasing interest in food products containing probiotic bacteria. The addition of probiotic bacteria as functional food supplements has become popular due to the health benefits of these bacteria [1,2,3,4]. The probiotics segment dominates the functional food ingredients market. Evidence from scientific studies suggests that probiotic strains exert a beneficial effect against various disorders, such as gastrointestinal diseases, bacterial vaginosis and urinary tract infections [5,6]. Probiotics have been implicated in inhibiting enteric pathogens, maintaining gut permeability, modulating the immune system, reducing inflammation, alleviating lactose intolerance, enhancing bowel motility and reducing cholesterol concentration [7,8,9]. To confer these health benefits to the host, a sufficient number of live cells is required to adhere to the host colon. As defined by the Food and Agriculture Organization of the United Nations and the World Health Organization (2002), probiotics are living microorganisms which, when administered in sufficient amounts, confer health benefits to the host. However, the viability of probiotic bacteria is questionable when they are exposed to harsh environments during processing (i.e., dehydration), storage and delivery to their site of action (i.e., the gastrointestinal tract [GIT]) [10].

Several studies have reported that oral doses higher than 10^9^ colony-forming units (CFUs) per day are required to restore and maintain the balance of bacteria [11,12]. Thus, probiotic bacteria should maintain high levels of viability during processing and remain alive during storage and delivery; for example, as they pass through the GIT. The survivability and dose levels of probiotics during storage and delivery are important parameters for probiotic efficacy. During storage and delivery by oral administration, probiotics are exposed to water, oxygen, heat, strong acid and bile [13,14]. To overcome these adverse factors, various dosage forms, such as capsules, tablets, powders and liquids, have been used [15,16,17,18,19]. Moreover, some special forms, such as vaginal suppositories and eye drops are also used [20,21,22]. These forms were first designed to maintain the viability of the probiotic bacteria during storage and delivery. With the development of new technologies, such as materials and embedding technologies, targeted release and directional delivery have become important research directions.

In this review, the characteristics of different dosage forms of probiotic products are first discussed for the treatment of different disorders and for different probiotic release sites and then some key factors affecting the efficiency of probiotic delivery are explored. Finally, innovative research on probiotic delivery systems in recent years is presented. This review aims to provide insights into how to choose the most appropriate dosage form for probiotic administration.

## 2. Dosage Forms Containing Probiotics

### 2.1. Liquids

The first probiotic products available were mostly liquid formulations [23]. Probiotics in liquid form are commercially available in various food matrices. Fresh dairy products are the most common products used for probiotic delivery [24]. During the past few years, the diversity of probiotic foods on the market has increased. Probiotics can now be found in non-fermented milk, fruit and berry juices and cereal-based products [25]. As an example, a functional probiotic drink using rice and soy as the fermentation substrates produces unique flavor substances and bioactive substances through the combination of the two ingredients, which can be developed as a new type of plant-based drink [26]. Although there is relatively little published information on the survival of probiotics in non-fermented food matrices, probiotic bacteria in yogurt or fruit products generally show low viability after storage and oral administration [24,27]. In fruit drinks, the characteristics of the fruit, especially the acidity, are key factors in maintaining the viability of the probiotics [28]. In the probiotic yogurt drinks, the protein stability and fermentative viability of the yogurt can be enhanced by the addition of a combination of prebiotics and hydrocolloid stabilizers [29]. It is worth emphasizing that the choice of different prebiotics affects the rheological properties and sensory characteristics of yogurt such as acidity, viscosity, firmness and syneresis during the storage period, therefore, new fermented milk products can be developed with combinations of prebiotics and probiotics to obtain new taste profiles [30].

### 2.2. Powder

The low transport and storage temperature requirements are the main commercial disadvantage of these liquid preparations, since the environmental stresses such as pH, water activity and oxygen in liquid preparations can affect the viability of probiotics [31]. To minimize the costs, it is important to produce probiotics in dry form. Orally administered powders are dry, solid granules made from a homogeneous mixture of a drug and its excipients. Probiotic powder, in the form of a dry powder, exhibits various advantages, such as convenient handling, storage and transportation and it can be used individually as a dosage form or as an intermediate in many other probiotic dosage forms [32]. Manufacturing dehydrated probiotic powders is challenging because it involves maintaining a large bacterial population and high viability after dehydration to prolong the storage time in complex environments [33]. Encapsulation techniques enable the preservation of microbial bioactivity through the use of protective materials, in addition to a controlled release and optimized delivery to ensure that probiotics are delivered to the specific site of action [34]. In a previous study, *Lactobacillus acidophilus* was encapsulated in 20% maltodextrin and different concentrations of gum arabic by spray drying and the effects of the gum arabic concentration and inlet temperature on the water activity, encapsulation efficiency and hygroscopicity of the probiotic powder were investigated. It was found that gum arabic and maltodextrin were structurally stable during spray drying and that the encapsulated probiotic cells had higher levels of viability in the simulated gastric fluid [35].

### 2.3. Capsules

Delivering viable probiotic cells to the GIT is challenging, especially when the probiotic product is in liquid or powder forms. Capsules are considered to be one of the ways to address this challenge. Capsules are solid dosage forms with either a hard or soft soluble container or shell made of a suitable form of gelatin [36]. Most dietary supplements are sold in the form of capsules. Hard capsules are generally preferred to administer probiotics. These capsules are available in different sizes and varieties and contain probiotics in the form of powders or microcapsules. They may also contain excipients, such as diluents, glidants, disintegrants or fillers [37]. These excipients maintain the physiology of the selected probiotic. The capsule shell protects the bacterial core from the acidic environment of the stomach and avoids the deleterious effects of bile salts. By changing the material of the shell, capsules can help with the delivery or control the release of the bacterial core at a desired site in the GIT [38,39,40].

A number of patented capsule technologies, such as DRcaps^®^, Vcaps^®^ and Vcaps^®^ Plus, have been developed in recent years to improve the capsule delivery systems by shortening or lengthening the release time using a variety of means, including modified polymeric carriers, thus allowing probiotics to reach specific locations [41]. DRcaps^TM^ show a dissolution profile that is resistant to a low pH. Marzorati et al. demonstrated that compared with hard gelatin capsules and Vcaps^®^ formulations, DRcaps^TM^ show an increased ability to protect probiotic microorganisms (an increase of at least 1 log) during gastrointestinal digestion and show 100% residual viability of probiotic bacteria in a 24-month shelf-life test [39].

### 2.4. Microcapsules

Microcapsules are very small capsules containing a material (such as an adhesive or a medicine) that is released when the capsule is broken, melted or dissolved [42]. They range from nanometers to millimeters in diameter [43]. The microencapsulation of probiotic bacteria is a promising technology to ensure bacterial stability during the drying process and to preserve their viability during storage, without a significant loss of functional properties, such acid tolerance, bile tolerance, surface hydrophobicity and enzyme activities [44]. The encapsulation of probiotics is used to increase the resistance of bacteria to freezing and freeze drying. In most studies, probiotic bacteria have been entrapped in a gel matrix of biological materials, such as alginate, β-carrageenan and gellan/xanthan [45]. The core and wall solution are converted to drops of a desired size using an extrusion method, an emulsion or by transfer from organic solvents. Dried probiotic microcapsules can be coated with an additional layer (shell) to protect the bacterial core from the acidic environment of the stomach and to avoid the deleterious effects of bile salts on the bacterial cell membrane. This additional shell can help release the bacterial core at a desired site in the GIT [46].

### 2.5. Tablets

The tablet, a dosage form with a high share of the global market, provides many advantages, such as physicochemical stability, a simple manufacturing process, low manufacturing cost and a high level of acceptance by consumers [47]. Although tablets are not the preferred dosage form for probiotic preparations, the properties of tablets make them an important direction for probiotic drug development. In view of the adverse effects on the bioactivity of probiotics caused by compression and wet granulation methods, the general process for probiotic tablet formulation is to mix the powder with an excipient after a drying procedure and then press the tablets directly into shape [48]. However, processes such as drying, mixing and compression inevitably destroy a wide range of cellular and biologically active components of probiotics, which is a challenge that needs to be addressed in the design of probiotic tablets [49].

### 2.6. Suppositories

A suppository is a solid drug delivery system that typically dissolves and releases its components at normal body temperature [50]. These delivery systems include rectal, vaginal and urethral suppositories. Suppositories are capable of preserving probiotic viability to a considerable degree and they are suitable for mass production and molding [21]. The vast majority of probiotic suppositories are vaginal suppositories, which maintain dosage uniformity, can provide less irritation to the vagina than other forms, such as effervescent tablets, and eliminate the need for large amounts of solution to dissolve the drug, such that it is more likely to be accepted by the user [51].

A complex microbiota exists in the female reproductive tract, with *Lactobacillus* as the dominant bacterial genus playing a positive role in preventing the invasion of pathogenic bacteria and regulating the ecological balance of microorganisms in the vagina. Thus, *Lactobacillus* abundance can be used as a bioindicator of vaginal health [52]. Probiotic vaginal suppositories help regulate the vaginal microbiota when an imbalance of *Lactobacillus* bacteria leads to an imbalance of other microbiota, which often induces various vaginal disorders. Some studies have shown that *Lactobacillus* vaginal suppositories are better colonized compared with oral probiotic preparations [53]. It has also been shown that the combination of probiotic vaginal suppositories and antibiotics is helpful for patients with repeated implantation failure [54].

## 3. The Site of Action of Probiotics and their Corresponding Dosage Forms

Probiotics come in a variety of dosage forms, but there are only two common forms, namely, oral and topical. Nasal sprays act on the nasal mucosa, suppositories are released directly into the vaginal or the rectum environment and other types of dosage forms enter the body through the oral route. We will describe this in more detail below.

### 3.1. Oral Cavity

Due to the unique biological functions of the human oral cavity, the resident microbial community is a complex system, consisting of more than 700 species [55]. The distribution of oral microbial populations is closely related to personal dietary habits, hygiene habits and immunity and it evolves with the development of metabolic networks formed by microbial interactions [56,57]. Biofilms are observed on some tooth surfaces, enamel and other parts of the oral cavity due to the adhesion, aggregation and colonization of various bacteria [58,59]. When the microbial community in the oral cavity is imbalanced, various oral diseases may occur, which in turn, affects the health of different parts of the body [60].

Dental caries, periodontal inflammation and oral candidiasis are three common oral diseases that have been reported to be positively affected by probiotics [61,62,63]. A randomized, double-blind trial demonstrated that the short-term daily consumption of a probiotic combination of *Lacticaseibacillus rhamnosus* GG and *Bifidobacterium animalis subsp. lactis* BB-12 reduces the abundance of *Plaque actinomycetes* and *Pseudomonas gingivalis* and promotes gum health in adolescents [64]. The mechanism of action of probiotics in the oral cavity is not entirely clear, but many studies have reported their capacity to modulate the inflammatory response, produce beneficial metabolites, such as bacteriocins and lactic acid, and compete with pathogenic bacteria for adhesion to biofilms on oral surfaces, which decreases the number of pathogenic bacteria in the oral cavity [59]. As an example, *Streptococcus salivarius* K12 is an oral probiotic that has been commercially developed [65]. *S. salivarius* K12 is known to regulate the immune properties of epithelial cells, as determined by microarray-based assays and enzyme-linked immunosorbent assays, while its production of the bacteriocin inhibitors, salivin A and salivin B, has been shown to have an inhibitory effect on *Streptococcus pyogenes* in vitro [66].

A number of probiotic dosage forms have now been developed for the treatment of oral diseases, of which probiotic orodispersible tablets (ODTs) represent a new pharmaceutical trend [67]. After consumers have taken ODTs, the tablet contents are released within seconds of contact with saliva and remain in areas such as the gums and oral mucosa to exert their positive effects. The combination of mucoadhesive polymers and probiotics increases the adhesive properties of ODTs and improves their retention of probiotics in the oral cavity [68].

### 3.2. Upper Respiratory Tract

The nasal cavity is part of the human respiratory tract and is an essential interface at which the body comes into contact with gases, pollutants, microorganisms, allergens and other substances from the external environment. The nasal cavity has a complex and diverse microbial community [69]. The microbiota in the healthy nasal cavity acts as an immune barrier against foreign infectious agents, while epithelial cells proliferate and the abundance of pathogenic microorganisms in the nasal cavity increases during the inflammatory response [70]. The association between dysbiosis of the nasal microbiome and respiratory diseases, such as allergic rhinitis (AR), chronic rhinosinusitis, otitis media and asthma, is also of increasing interest due to the large number of patients affected by respiratory diseases [71].

For these upper respiratory diseases, oral probiotic preparations may not be particularly effective. A randomized controlled trial of early probiotic supplementation in infants showed a reduction in the incidence of asthma after oral probiotic supplementation, but without statistical significance [72]. For respiratory allergic conditions such as AR, oral supplementation with probiotics shows very limited effectiveness [73]. Compared with oral administration, nasal probiotic drops or nasal sprays may be more beneficial. Animal trials have demonstrated that the administration of probiotic nasal drops containing *Bifidobacterium* and *Lactiplantibacillus plantarum* significantly alleviates ovalbumin-induced AR in mice by restoring the Th2/Treg cell balance and modulating the intestinal microbiota [74]. Similar results were found with nasal drops of probiotic extracts [75]. *S. salivarius*, the primary colonizing bacterial species in the nasopharyngeal microbiota, has been used in a nasal spray, with promising results for the treatment of acute otitis media [76,77]. Probiotic spray-dried biologics targeting the nasal cavity have been used as immune adjuvants to compete with pathogenic bacteria in the upper respiratory tract [78].

### 3.3. Gastrointestinal Tract

The intestine is the main site of action of probiotics and the development of various targeted delivery systems has enabled probiotics to reach their designated sites and exert beneficial effects. Despite the fact that the mechanism has not been fully studied, several studies have confirmed that the defense function of the intestinal barrier, which is composed of intestinal epithelial cells, is enhanced when probiotics reach the intestine [79,80,81]. Adhesion of probiotics to the intestinal mucosa also increases, which is a favorable factor for the interaction between probiotic bacteria and the host during intestinal colonization [82]. Dysbiosis of the intestinal microbiota community structure is correlated with several intestinal diseases and metabolic disorders, such as inflammatory bowel disease, constipation, diabetes and obesity [83,84,85]. Probiotics also have positive implications in the alleviation of diseases caused by intestinal microbial imbalances. In this regard, probiotics have been widely accepted as a good option for regulating the intestinal microecological balance.

The occurrence of gastrointestinal diseases, such as irritable bowel syndrome, inflammatory bowel disease and intractable constipation, have been shown to be strongly correlated with intestinal microbiota activity. A meta-analysis of the efficacy of probiotic preparations in premature children with necrotizing colitis showed that both *Lactobacillus* and *Bifidobacterium* reduced the risk of necrotizing enterocolitis compared with the placebo group, but there was no significant heterogeneity in the relationship between probiotics and mortality [86]. Another randomized double-blind controlled trial investigated the clinical efficacy of a multi-strain probiotic product on the gut microbiological profile of patients with functional constipation. Although there were no significant effects on the symptoms, the probiotics helped regulate gut function and relieve constipation earlier compared with the placebo control group [87].

Metabolic syndrome is a pathological condition in which multiple metabolic components accumulate abnormally. It is characterized by hypertension, dysglycemia and dyslipidemia and it has received attention as an important health problem [88]. The regulation of the intestinal microbiota by probiotics leads to a reduction in insulin resistance and blood glucose levels [89,90,91]. Probiotics also reduce oxidative stress and uric acid levels, in addition to improving insulin sensitivity [92]. As intestinal microorganisms affect the central nervous system through the gut-brain axis, the effect of probiotics on metabolism-related symptoms in psychiatric disorders has also received attention. Probiotics have been shown to alleviate metabolic disorders and cognitive impairment in patients with schizophrenia [93,94]. Probiotics are gaining attention as a novel form of therapy for metabolic diseases.

### 3.4. Vagina

Aerobic and anaerobic micromicrobiota coexist in the female reproductive tract in a dynamic equilibrium and their community composition may be affected by a variety of factors, including age, endocrinology and sexuality [95]. *Lactobacillus* is the dominant bacterial genus in the community, helping to maintain microbial homeostasis in the vagina [96]. When the abundance of *Lactobacillus* decreases, the resulting increase in anaerobic and pathogenic bacteria in the micromicrobiota results in an unbalanced vaginal microbiota, causing bacterial vaginosis (BV) [97]. BV is treated most frequently with antibiotics, but the misuse of antibiotics can lead to the accumulation and spread of antibiotic-resistant genes, leading to the development of drug-resistant bacteria [98]. Probiotics have emerged as a new treatment method for BV because of their ability to stimulate and strengthen the immune system and inhibit the proliferation of pathogenic bacteria [99].

A new bilayer vaginal tablet of *Lactococcus lactis* has been designed for the treatment of vaginal bacterial infections. The effervescent layer of the tablet is released rapidly, while the matrix layer is released slowly over 24 h. After 3 months of stability studies, in which the tablets were placed at room temperature and in a desiccator, the bacteria were found to be retained at approximately 10^8^ CFU/g, achieving the desired drug properties. Thus, this tablet is promising for use in the treatment of vaginal diseases [100].

The effect of a yogurt drink spiked with a *Lactobacillus* strain was evaluated in a double-blind, randomized, controlled clinical trial of 36 women with BV. The women consumed the yogurt drink or a placebo twice daily for 4 weeks, accompanied by 1 week of antibiotic treatment. After 4 weeks of intervention, none of the seventeen participants in the *Lactobacillus* group had BV, compared to six of the seventeen participants in the placebo group, indicating that probiotic strains significantly increase recovery from BV [101]. However, antibiotics were also used in this experiment and the effect of the oral probiotics alone on regulating the vaginal microbiota could not be ascertained. To demonstrate that probiotic preparations can be an alternative to antibiotics, another study with 554 women of an appropriate age demonstrated that oral capsules of probiotics are also effective and can restore the balance of the vaginal microbiota [102].

## 4. Factors Affecting Probiotic Survival

### 4.1. Processing Technology

After dehydration, the vital metabolic activity of an organism or cell is extremely reduced and it is maintained in a state in which vital functions have almost ceased [103]. The drying process affects the properties of probiotics, such as cell surface hydrophobicity, tolerance to environmental stresses and antimicrobial activity [104]. The survival time of the probiotic depends mainly on the drying technique and the storage method. Drying methods for microorganisms include freeze drying, spray drying, vacuum drying and fluidized-bed drying [105]. Using a combination of optimizing protective agents, a suitable drying method with optimum setting conditions and the selection and characterization of appropriate strains, the viability of probiotic bacteria can be maintained at a high level during processing [106]. However, some methods cannot be used in large-scale industrial processes because of their high cost. Here, we mainly introduce two types of drying techniques, spray drying and freeze drying.

Spray drying can be divided into four stages: atomization, gas heating, particle formation and separation [107,108]. During the spray drying process, the probiotic bacteria are affected by desiccation, heat, oxidation and osmotic stresses and their cell membranes are easily damaged, leading to their death [109]. Therefore, it is extremely important to choose protective material types and control process parameters, such as temperature, time and the feed rate, during the entire process [110]. Freeze drying mainly includes three important steps: freezing, primary drying and secondary drying [111]. Compared with spray drying, the strain is protected from thermal damage and oxidative stress during freeze drying [112,113]. To reduce the damage to the bacterial structure and adverse effects on probiotic properties during freeze-drying engineering, sugar cryoprotectants, such as maltodextrin and sucrose, are generally added [114].

Many studies have compared spray-drying and freeze-drying methods for probiotics. For example, one study compared the effects of spray drying and freeze drying on the survival of *Lacticaseibacillus casei* Shirota under different stress conditions. The results showed that spray-dried microcapsules had a smaller decrease in cell viability in an acidic environment than freeze-dried microcapsules. However, freeze-dried microcapsules were more protective of probiotic cells at 85 °C and 90 °C. The cell viability of both microcapsules decreased by approximately 2.5 log after exposure to 3% bile salts, with no significant difference between the two microcapsules [115]. In an additional study, the relationship between the accumulation of osmoprotectants and the stress tolerance of *Propionibacterium freudenreichii* was monitored by adjusting the growth conditions [116]. The results showed that the accumulation of trehalose correlated with the survival of the bacteria after spray drying, while the accumulation of glycine betaine was associated with the survival of the strain after freeze drying. These findings can be used as a reference for optimizing the drying process of probiotics.

Compression is another process that can easily damage probiotics. Direct compression is regarded as the method of choice for manufacturing tablets with inhaled and moisture-sensitive active ingredients for industrial use [117]. As direct compression inevitably causes damage to bacterial morphology, it is essential to investigate the relationship between compression force and probiotic cell viability. One study showed that as the concentration of hypromellose phthalate increases, tablets made with high tensile and compressive strengths exhibit a slow release rate and greater than 80% bacterial cell viability [118]. Meanwhile, another study reported that when the cell density of the tablets increases, the particle gap is too small and high levels of mechanical stress may cause cell rupture and thus reduce the survival of probiotic bacteria [119]. The difference between these two results may be attributed to variations in drying processes and excipients. Notably, the species of the strain also affects the sensitivity of directly compressed probiotic tablets, as some strains have cell surface molecules, such as exopolysaccharides, that reduce cell damage during compression [120]. Due to strain specificity and the variability in excipients, the appropriate choice of compression force during the manufacturing of probiotic tablets can substantially improve strain survival.

### 4.2. Storage Conditions

Storage is an important aspect of probiotic preparations before they are used, as storage conditions directly affect the biological viability and effectiveness of the preparation. Factors such as temperature, water activity, oxygen content, composition of the probiotic preparation, storage time and pH level are all crucial during the storage process (Figure 1). Probiotics are extremely temperature-sensitive, so they are generally stored at 4 °C, as room temperature storage shortens their shelf life. It has been shown that probiotics, especially anaerobic bacteria, such as *Bifidobacterium bifidum*, are more viable and physiologically more functional in low-oxygen conditions and in low-water-activity matrices [121,122]. To mitigate the oxidative stress of probiotics in formulations, different oxygen scavengers, such as cysteine and ascorbic acid, have been developed and the oxygen transmission rate of packaging materials has been decreased [123].

## 5. Solutions to Increase the Viability of Probiotics

### 5.1. Common Multifunctional Polymeric Materials

Proteins have become a great option for encapsulating probiotics. Due to their biodegradability, biocompatibility and non-toxicity, proteins may be used to form natural hydrogels for oral delivery using enzyme cross-linking, thermally controlled sol transition or chemical cross-linking [124]. Gelatin, whey protein and casein are some of the common types of proteins used for encapsulation. The use of soybean isolate for the preparation of probiotic particles has also been reported [125]. However, during gastric transit, digestive enzymes cause the rapid degradation of protein gels and thus decrease their bioavailability. Therefore, proteins are often used in combination with polysaccharide molecules and lipid compounds.

Polysaccharide compounds are one of the main groups of polymers used for the targeted delivery of probiotics [126]. The polysaccharide macromolecules remain stable in the gastric environment and protect the probiotics from the acidic environment of the stomach. Various hydrolytic enzymes present in the colon, such as β-D-galactosidase and β-D-glucanase, break the glycosidic bonds of the polysaccharides to prepare them for degradation and induce the release of the probiotics [124]. The polysaccharides frequently used in research are alginate, chitosan, extracellular polysaccharide and carboxymethyl cellulose. Sodium alginate is widely used due to its safety, acid-base sensitivity and cost-effectiveness [127]. To maintain the stability of sodium alginate, it is often used in conjunction with other polysaccharides or proteins to achieve the greater protection of the probiotics.

Biodegradable hydrogels are cross-linked polymer networks that can maintain a certain amount of water without being dissolved, thus protecting probiotics from the harsh external environment and reliably delivering probiotics to the colon in a targeted manner [128]. A study was conducted in which two secondary polysaccharides, low-methoxyl pectin and K-carrageenan, were mixed with sodium alginate to form dual-network hydrogel particles. The physical properties, cross-linking ability, swelling coefficient and strain survival of the hydrogel particles with different mixing concentrations were then compared. The results showed that all of the polysaccharides could be cross-linked to prepare hydrogel particles containing probiotics and that the combination of alginate and low-methoxyl pectin enhanced the structural stability of the particles to a certain extent [129].

As polysaccharides are mostly water-soluble, unmodified polysaccharides may be released prematurely, reducing the number of probiotics delivered to the colonic site [130]. Several recent studies have demonstrated the potential of modified polysaccharides in multifunctional carriers (Table 1). For example, the use of epigallocatechin-3-gallate-modified succinate-grafted alginate was shown to improve the thermal stability, viscosity properties, emulsion stability and viability of the tested strains [131].

### 5.2. Nanocarriers

The presence of edible nanomaterials has no toxic effect on the food matrix, improves the bioavailability of bioactive components in the food and enhances the compatibility of the components in the matrix [142]. Moreover, the pore size and diameter of nanostructured particles can be controlled to decrease their release rate in the harsh environment of the GIT and thus, they are an option for constructing targeted probiotic delivery systems [143]. There are two typical uses of nanomaterials in probiotic dosage forms. They can be used as carriers to deliver probiotics and as excipients to enhance probiotic survival and bioactivity.

Electrospinning technology uses electrostatic forces to produce nanofibers by manipulating the charged filaments of polymer solutions or melts [144]. Nanofibers exhibit excellent potential for drug encapsulation due in part to their high loading capacity and outstanding controlled release properties [145]. Bacterial cellulose and fructose have been made into nanofibers using the electrostatic spinning technique to explore their feasibility for probiotic encapsulation [146,147]. Nanofibers made of bacterial cellulose can remain stable at 180 °C and *L. acidophilus* 016 immobilized in these nanofibers has a survival rate of 71.1% after 24 days of storage at room temperature [147]. Likewise, electrospun fibers made from fructose and polyvinyl alcohol and used as wall materials enhance the viability of probiotic bacteria and are stable under humid conditions at 70 °C [146]. Recently, an innovative study used a synthetic polymer to form a thick nano-coating that self-assembles on *Escherichia coli* Nissle 1917 and intelligently releases the bacterium in response to intestinal signals [148]. Eudragit L100-55, the nanomaterial used in that study, is an anionic polymer based on methacrylic acid and ethyl acrylate that dissolves only at pH > 5.5 and shows good enterosolubility [149].

### 5.3. Lipid Membranes and Biofilms

Simple and rapid methods of protecting probiotics are now emerging, allowing the rapid production of probiotic preparations in a fraction of the time, while significantly protecting their biological activity in extreme environments. In a previous study, a mixture of dimethylbenzoic acid and cholesterol was vortexed with a variety of bacteria, including *E. coli* and *Staphylococcus aureus*, to self-assemble lipid membranes onto the bacterial surface [150]. The bioactivity of the bacteria was unchanged after self-assembly, while their intestinal bioavailability increased fourfold, demonstrating the high efficiency of supramolecular self-assembly-coated bacteria and the potential of this probiotic delivery system.

Biofilm formation can occur through a variety of pathways that are regulated by extracellular matrix components and are related to environmental factors and bacterial community responses [151]. The dual functions of biofilms to enable physical adhesion and act as a defensive barrier have enabled them to become a novel choice for probiotic coatings. Coating a *Bacillus subtilis* biofilm onto the surface of probiotics increases their bioavailability by 125-fold, demonstrating the potential of bacterial biofilms for gastrointestinal delivery [152].

## 6. Conclusions

Probiotics are used to improve the micro-ecological balance in various parts of the body, particularly in the GIT. Due to their outstanding properties and clinical value, probiotics are currently administered as dietary supplements and as usable food ingredients. In this article, we reviewed the characteristics of different dosage forms of probiotic preparations and their modes of action, with particular reference to their various clinical applications. We then described the various challenges encountered during the production, storage and in vivo transport of probiotics. Finally, we outlined some ways to improve the bioavailability of probiotic preparations.

Currently, most research attention is focused on the development of encapsulation materials for probiotic preparations and the functional characterization of probiotics. For a comprehensive understanding of the safety and effectiveness of probiotic preparations, the safety of the encapsulating material, the dosage of probiotics used, the feeding time and other related factors should be considered together, especially when the preparations are used for clinical treatment. It is also necessary to update and develop relevant policies and regulations for the production of probiotic functional foods and drugs to prevent consumers from being deceived by false claims.

## Figures and Tables

**Figure 1 foods-11-02472-f001:**
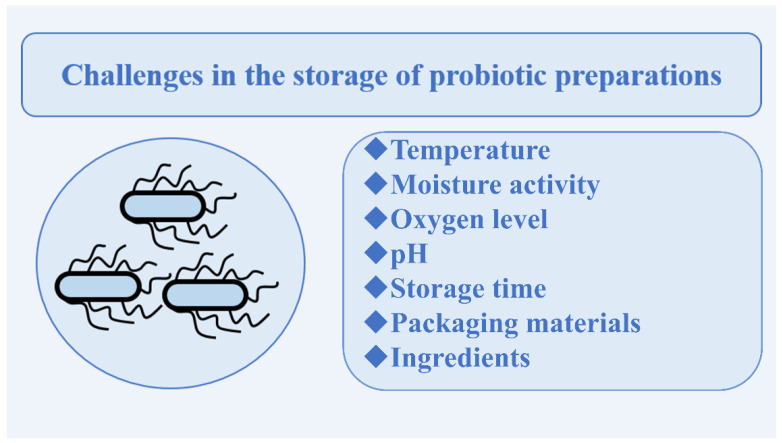
Factors affecting the storage stability of probiotic preparations.

**Table 1 foods-11-02472-t001:** Recent studies of different carriers for the targeted delivery of probiotics.

	Polymer	Form	Survival Rate and Stability of the Strains	Reference
*Lacticaseibacillus paracasei* KS-199	Alginate	Electrospun nanofiber	The survival rate of the strain after electrostatic spinning was 85.87%. Under simulated gastrointestinal conditions, the survival rate of the encapsulated strain increased from 51.8% to 70.8%.	[132]
*L. plantarum*	Ca-alginate and chitosan	Microcapsule	The viability of the strains mixed with inulin was 7.23 ± 0.21 and 9.15 ± 0.33 log CFU/g at 25 °C and 4 °C storage, respectively. The viability of the strains mixed with inulin or resistant starch after 90 days of storage was 7.37 ± 0.12 and 7.82 ± 0.39 log CFU/g, respectively.	[133]
*Limosilactobacillus reuteri*	Type-A gelatin/sodium caseinate (GE/Cas), type-A gelatin/gum arabic (GE/GA)	Microcapsule	The survival of the strains decreased in the order GE/Cas > Cas > GE/GA > GE after simulated digestion, heating and ambient storage.	[134]
*Bifidobacterium animalis* subsp. *lactis* BLC1	Proanthocyanidin-rich cinnamon extract (PRCE)	Microparticle	The encapsulation rate of probiotics with the combination of BLC1 and 5% PRCE was 98.59 ± 0.45%. After 120 days of storage at 7 °C, the viability of BLC1 was 9.30 ± 0.16 log CFU/g.	[135]
*L.paracasei* LS14	Soy protein isolate (SPI) and sugar beet pectin (SBP)	Hydrogel	The survival rate of probiotics encapsulated in an SPI/SBP hydrogel in simulated gastric juice was greater than 96.4%. The greatest storage stability was seen for the probiotic wrapped in an interpenetrating polymer network hydrogel containing 10% SPI, 3.5% SBP and 10 U laccase.	[128]
*L. acidophilus* La-14	Calcium alginate, whey proteins and sodium alginate	Microparticle	The multilayer calcium alginate particles were encapsulated with greater than 80% efficiency and had high strain viability when exposed to simulated gastrointestinal and thermal treatment conditions. The combination of whey protein and one layer of sodium alginate coating was optimal.	[136]
*L.rhamnosus*	Hyaluronic acid	Microcapsule	The hydrogel was most stable at a concentration of 4% (*w*/*v*). The viability under a simulated gastrointestinal tract and the storage stability of the strains were enhanced after microencapsulation.	[137]
*Kluyveromyces marxianus* VM004	Whey protein concentrate (WPC) and water-soluble chitosan (WSCh)	Microcapsule	After spray drying, the probiotic powder had a viability of 8.38 log CFU/g. At 30% (*w*/*v*) solids (29:1 WPC:WSCh), the survival rate of the strain after a gastrointestinal tolerance test was up to 95%.	[138]
*L. rhamnosus* ATCC 7469	Whey protein isolate, crystalline nanocellulose and inulin	Microcapsule	The probiotic bacteria encapsulated in the microcapsules remained active up to 3.2 × 10^5^ CFU/g after being exposed to simulated gastric fluid at 37 °C for 60 min and then exposed to 0.6% bile salt at pH 7.34 for 120 min.	[139]
*Saccharomyces boulardii* ATCC MYA-796	Alginate and alginate–chitosan	Microcapsule	The survival rates of alginate- and alginate–chitosan-microencapsulated yeast were 80% and 90% after 240 h of treatment with simulated gastric fluid and 80% and 85% after 240 h of treatment with simulated intestinal fluid.	[140]
*Saccharomyces cerevisiae* JCM 7255	Alginate and skim milk	Microcapsule	The survival rates of encapsulated yeast under simulated gastric and bile conditions were significantly higher and remained high after 14 days of storage at 25 °C.	[141]

## Data Availability

No new data were created or analyzed in this study. Data sharing is not applicable to this article.

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
