# Peer review of "Characteristics of Probiotic Preparations and Their Applications"

_foods, 2022, doi:10.3390/foods11162472_

Round 1

Reviewer 1 Report

Wang et al. submitted their manuscript entitled “Characteristics of probiotic preparations and their applications“ to Foods. The manuscript is interesting and well written. Thus, I have a few comments only.

In the abstract, it would be suitable to discriminate Background, Scope and approach, etc. From other texts. It should be in bold, italics, using the colon, or in another way. Now it has been merged with other text.

L77-81: This text should be used as a general introduction but not as a part of the paragraph dealing with probiotics in liquids.

84-85: "Compared ..." - This sentence is excessive. Please, remove it and rephrase the text.

L100-101: What do you mean by activity? Viability?

L166-167: The term flora was replaced by microbiota many years ago. Please, do not use flora if your manuscript is not dealing with flowers and trees.

L174-177: I do not think that probiotic drinks have a relation to the reproductive tract. Please, rephrase the sentence.

L193: Bifidobacterium animalis subsp. lactis BB-12. Please, precise it.

L256: There are not premature necrotizing colitis but premature children. Please, rephrase the sentence

L295-296: Yes, there is a relation between probiotics and the reproductive tract. However, the use of antibiotics in this study is problematic as it is written.

Reviewer 2 Report

Dear Editors and authors,

The manuscript (Characteristics of probiotic preparations and their applications) has good idea and modern topic but it needs some corrects and modifications.

1-  The abstract of the manuscript does not reflect the topics of the manuscript and does not contain a clear conclusion, it must be rewritten again.

2- The writing style of the references is not match with the references style of foods journal.

3-The manuscript needs to be strengthened by adding a number of references because there are several paragraphs without any reference. I suggested adding how many reference, see Page 3 line 140 (I suggest you to add new reference in here (Niamah, A. K., Al-Sahlany, S. T. G., Ibrahim, S. A., Verma, D. K., Thakur, M., Singh, S., ... & Utama, G. L. (2021). Electro-hydrodynamic processing for encapsulation of probiotics: A review on recent trends, technological development, challenges and future prospect. Food Bioscience, 44, 101458.‏)

Page 4 line 149 Add new reference in here (Vorländer, K., Kampen, I., Finke, J. H., & Kwade, A. (2020). Along the process chain to probiotic tablets: evaluation of mechanical impacts on microbial viability. Pharmaceutics, 12(1), 66.‏)

Page 7 line 311, Rajam, R., & Subramanian, P. (2022). Encapsulation of probiotics: past, present and future. Beni-Suef University Journal of Basic and Applied Sciences, 11(1), 1-18.‏

4-The names of lactic acid bacteria should be written according to the modern nomenclature such as Lactobacillus casei correct to Lacticaseibacillus casei.

5-Figure 1. Factors affecting the storage stability of probiotic preparations, I suggest adding that storage time) is an impact factor on the viability of probiotic bacteria in products.

6-Table 1 page 9, The table needs some references about encapsulation of probiotic yeast.  In order to improve the quality of the table and to be clear to readers , I suggest you

1-Niamah, A. K., Al-Manhel, A. J., & Al-Sahlany, S. T. G. (2018). EFFECT MICROENCAPSULATION OF Saccharomyces boulardii ON VIABILITY OF YEAST IN VITRO AND ICE CREAM. Carpathian Journal of Food Science & Technology, 10(3): 100-107. 

 2-Pinpimai, K., Rodkhum, C., Chansue, N., Katagiri, T., Maita, M., & Pirarat, N. (2015). The study on the candidate probiotic properties of encapsulated yeast, Saccharomyces cerevisiae JCM 7255, in Nile Tilapia (Oreochromis niloticus). Research in Veterinary Science, 102, 103-111.

7- Some abbreviations should be well explained to the reader such as GIT.

Round 2

Reviewer 2 Report

Dear Editors, 

The authors made all the necessary changes to improve the manuscript, and now I recommend it for publication in its current form.